J Physiol 601.17 (2023) pp 3961–3974

3961

# Higher gestational weight gain delays wound healing and reduces expression of long non-coding RNA *KLRK1-AS1* in neonatal endothelial progenitor cells

Elisa Weiss[1,2] , Anna Schrüfer[1] , Carolina Tocantins[1,3,4] , Mariana Simoes Diniz[1,3,4] ,
Boris Novakovic[5,6] , Anke S. van Bergen[7,8] , Azra Kulovic-Sissawo[1,2] , Richard Saffery[5,6] ,
Reinier A. Boon[7,8,9,10] and Ursula Hiden[1,2]

[1]*Perinatal Research Laboratory, Department of Obstetrics and Gynaecology, Medical University of Graz, Graz, Austria*
[2]*Research Unit Early Life Determinants (ELiD), Medical University of Graz, Graz, Austria*
[3]*CNC-Center for Neuroscience and Cell Biology, CIBB-Centre for Innovative Biomedicine and Biotechnology, University of Coimbra, Coimbra, Portugal*
[4]*PhD Programme in Experimental Biology and Biomedicine (PDBEB), Institute for Interdisciplinary Research (IIIUC), University of Coimbra, Coimbra, Portugal*
[5]*Molecular Immunity, Infection and Immunity Theme, Murdoch Children's Research Institute, Parkville, Victoria, Australia*
[6]*Department of Paediatrics, University of Melbourne, Parkville, Victoria, Australia*
[7]*Department of Physiology, Amsterdam University Medical Centers, Vrije Universiteit Amsterdam, Amsterdam, The Netherlands*
[8]*Amsterdam Cardiovascular Sciences, Microcirculation, Amsterdam, The Netherlands*
[9]*Institute for Cardiovascular Regeneration, Centre for Molecular Medicine, Goethe University Frankfurt am Main, Frankfurt am Main, Germany*
[10]*German Centre for Cardiovascular Research DZHK, Partner site Frankfurt Rhein/Main, Frankfurt am Main, Germany*

Handling Editors: Laura Bennet & Rebecca Simmons

The peer review history is available in the Supporting Information section of this article (https://doi.org/10.1113/JP284871#support-information-section).

R. A. Boon and U. Hiden are joint last authors.

The Journal of Physiology

**Abstract** High gestational weight gain (GWG) is a cardiovascular risk factor and may disturb neonatal endothelial function. Long non-coding RNAs (lncRNAs) regulate gene expression epigenetically and can modulate endothelial function. Endothelial colony forming cells (ECFCs), circulating endothelial precursors, are a recruitable source of endothelial cells and sustain endothelial function, vascular growth and repair. We here investigated whether higher GWG affects neonatal ECFC function and elucidated the role of lncRNAs herein. Wound healing of umbilical cord blood-derived ECFCs after pregnancies with GWG $<13$ kg *versus* $>13$ kg was determined in a scratch assay and based on monolayer impedance after electric wounding (electric cell-substrate impedance sensing, ECIS). LncRNA expression was analysed by RNA sequencing. The function of killer cell lectin-like receptor K1 antisense RNA (*KLRK1-AS1*) was investigated after siRNA-based knockdown. Closure of the scratch was delayed by 25% ($P = 0.041$) in the higher GWG group and correlated inversely with GWG ($R = -0.538$, $P = 0.012$) in all subjects ($n = 22$). Similarly, recovery of the monolayer barrier after electric wounding was delayed ($-11\%$ after 20 h; $P = 0.014$; $n = 15$). Several lncRNAs correlated with maternal GWG, the most significant one being *KLRK1-AS1* ($\log_2$ fold change $= -0.135$, $P < 0.001$, $n = 35$). *KLRK1-AS1* knockdown ($n = 4$) reduced barrier recovery after electric wounding by 21% ($P = 0.029$) and *KLRK1-AS1* expression correlated with the time required for wound healing for both scratch ($R = 0.447$, $P = 0.033$) and impedance-based assay ($R = 0.629$, $P = 0.014$). Higher GWG reduces wound healing of neonatal ECFCs, and lower levels of the lncRNA *KLRK1-AS1* may underlie this.

(Received 12 April 2023; accepted after revision 3 July 2023; first published online 20 July 2023)

**Corresponding author** U. Hiden: Department of Obstetrics and Gynaecology, Medical University of Graz Auenbruggerplatz 14, 8036 Graz, Austria. Email: ursula.hiden@medunigraz.at

**Abstract figure legend** Gestational weight gain (GWG) affects wound healing of fetal endothelial progenitor cells (endothelial colony forming cells, ECFCs). Cord blood-derived ECFCs were isolated after pregnancies with high *versus* low GWG, and wound healing capacity was compared. High GWG caused delayed wound healing of ECFCs. Correlation analysis and silencing experiments revealed that wound healing capacity was related with the expression of the long non-coding RNA (lncRNA) *KLRK1-AS1*.

## Key points

- Maternal cardiovascular risk factors such as diabetes, obesity and smoking in pregnancy disturb fetal endothelial function, and we here investigated whether also high gestational weight gain (GWG) has an impact on fetal endothelial cells.
- Circulating endothelial progenitor cells (endothelial colony forming cells, ECFCs) are highly abundant in the neonatal blood stream and serve as a circulating pool for vascular growth and repair.
- We revealed that higher GWG delays wound healing capacity of ECFCs *in vitro*.
- We identified the regulatory RNA lncRNA *KLRK1-AS1* as a link between GWG and delayed ECFC wound healing.
- Our data show that high GWG, independent of pre-pregnancy BMI, affects neonatal ECFC function.

**Elisa Weiss**' research interest is how influences during pregnancy shape the later health of children by epigenetic mechanisms. She completed her PhD studies in 2022 at the Department of Gynecology and Obstetrics at Med Uni Graz (Austria) in the research group of Ursula Hiden. Afterwards she started her PostDoc project there. Already in her studies, but also now as a PostDoc, she investigated specifically how maternal metabolic disturbances in pregnancy affect and disturb the function of fetal endothelial cells.

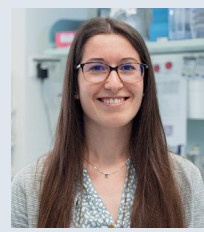

## Introduction

An intact endothelium is key to cardiovascular health, while endothelial dysfunction is associated with cardiovascular disease (CVD) (Little et al., 2021). Cardiovascular risk factors (CVRF), such as diabetes, smoking, obesity, unbalanced diet and sedentary lifestyle, disturb endothelial function (Jayedi et al., 2020) and foster the risk of developing CVD. In addition, CVRFs also include gestational weight gain (GWG). Notably, already a modest weight gain of 4 kg within 8 weeks causes impaired endothelial function in lean healthy adults, as indicated by decreased brachial artery flow-mediated dilatation (Romero-Corral et al., 2010). Evidence suggests that increased or excessive maternal weight gain in pregnancy also represents a CVRF as it raises the risk of developing hypertensive disorders (Galjaard et al., 2013; Yuan et al., 2022).

Using endothelial cells isolated from perinatal tissues such as umbilical cord and placenta, maternal CVRFs in pregnancy have been shown not only to affect maternal endothelial function, but also endothelial function of the fetus/neonate. For instance, *in vitro* experiments using feto-placental endothelial cells revealed that maternal gestational diabetes disrupts the actin cytoskeleton organisation with increased barrier integrity (Cvitic et al., 2018), and maternal obesity leads to endoplasmic reticulum stress in human umbilical vein endothelial cells (HUVECs; Villalobos-Labra et al., 2018). Interestingly, neonatal endothelial function is also affected by an abnormal GWG, independently from the pre-gestational body mass index (BMI): in pregnancies with normal maternal pre-pregnancy weight, excessive weight gain (>16 kg) reduces endothelial nitric oxide synthase protein expression in HUVECs (Pardo et al., 2015). Importantly, not only do severe metabolic changes in the mother affect neonatal endothelial cells, but also subtle alterations may have an effect: in feto-placental endothelial cells, the peptidase membrane-metalloendopeptidase (also termed neprilysin), which is involved in vascular tone regulation, is reduced by maternal pre-pregnancy overweight, starting with a BMI of >25 kg/m$^2$ (Weiss et al., 2020).

The intrauterine environment can influence the development and health of the offspring through epigenetic mechanisms, and non-coding RNAs, including short non-coding RNAs (<200 nucleotides, e.g. microRNAs) and long non-coding RNAs (lncRNAs, >200 nucleotides), are important epigenetic regulators (Kapranov et al., 2007). For example, a group of microRNAs termed angiomiRs modulate the expression of various genes affecting endothelial function (Boon et al., 2013; Cvitic et al., 2020; Demolli et al., 2015; Urbich et al., 2012). Recently, also lncRNAs have been found to regulate endothelial and vascular function (Hofmann et al., 2019; Jae & Dimmeler, 2020; Kremer, Bink et al., 2022; Michalik et al., 2014; Pham et al., 2020; Stanicek et al., 2020) and to be involved in the development of endothelial dysfunction and CVD, although detailed mechanisms are still lacking (Lozano-Vidal et al., 2019).

Endothelial colony forming cells (ECFCs) are a type of circulating endothelial progenitor cell that are recruited from the circulation to sustain vascular repair (Banno & Yoder, 2018) and angiogenesis (Asahara et al., 1997). Once extracted from the circulation, they give rise to mature endothelial cells (Paschalaki & Randi, 2018) and support the maintenance of proper endothelial function. In particular perinatally, ECFC numbers are high (Ingram et al., 2004), suggesting an important role of these cells in neonatal vascular remodelling, adaption and function. In a recent study, we have highlighted the great sensitivity of neonatal ECFCs to maternal influences: even in the healthy, non-diabetic range, a higher maternal fasting blood glucose is associated with slower *in vitro* ECFC colony outgrowth (Weiss et al., 2022).

In this study, we investigated the effect of GWG on the function of neonatal ECFCs, isolated from umbilical cord blood, and elucidated the potential role of the lncRNAs herein.

## Methods

### Ethical approval

The study conformed to the standards set by the *Declaration of Helsinki* (version 2013), except for registration in a database. It was approved by the ethics committee of the Medical University of Graz, Austria (29-319 ex 16/17). Only participants with written informed consent were included.

### Study cohort

Cord blood samples after delivery of singleton pregnancies were collected to isolate neonatal ECFCs ($n = 35$). Exclusion criteria included gestational diabetes mellitus (diagnosed by IADPSG criteria; International Association of Diabetes and Pregnancy Study Groups Consensus Panel, 2010), smoking (self-reported), medical disorders or pregnancy complications, use of any medication, and adverse medical history. To obtain similar sample size, a GWG of 13 kg was used as the cut-off for group comparison. Maternal and offspring main characteristics depending on GWG (<13 kg *versus* >13 kg) are presented in Table 1.

All 35 ECFC isolations were used for RNA sequencing. However, for scratch assay and electrical wounding (electric cell-substrate impedance sensing, ECIS) only

**Table 1. Maternal and offspring main characteristics of ECFC isolations**

| Characteristic | GWG <13 kg | GWG >13 kg | P |
|---|---|---|---|
| Number of subjects | 15 | 20 | |
| Neonatal sex (male/female) | 9/6 | 10/10 | 0.734 |
| Maternal age (years) | $30.2 \pm 5.7$ | $31.6 \pm 4.5$ | 0.425 |
| Pre-pregnancy BMI (kg/m$^2$) | 25.1 (21.7, 28.1) | 22.9 (21.3, 26.9) | 0.559 |
| BMI at delivery (kg/m$^2$) | $28.6 \pm 4.0$ | $30.8 \pm 4.7$ | 0.159 |
| Gestational weight gain (kg) | 10.0 (4.0, 11.0) | 16.0 (14.1, 20.1) | **<0.0001** |
| oGTT 0 h (mg/dl) | $81.2 \pm 5.6$ | $80.2 \pm 7.7$ | 0.672 |
| oGTT 1 h (mg/dl) | $123.1 \pm 30.3$ | $115.4 \pm 27.4$ | 0.437 |
| oGTT 2 h (mg/dl) | $106.5 \pm 20.3$ | $94.2 \pm 20.5$ | 0.090 |
| Gestational age at delivery (week) | $39.3 \pm 1.0$ | $39.4 \pm 0.8$ | 0.664 |
| Mode of delivery (vaginal/C-section) | 4/11 | 7/13 | 0.721 |
| Neonatal weight (g) | $3233.4 \pm 356.6$ | $3624.5 \pm 370.8$ | **0.004** |
| Neonatal height (cm) | $49.9 \pm 1.6$ | $51.9 \pm 2.5$ | **0.012** |
| Placental weight (g) | 560 (510, 670) | 700 (600, 850) | **0.013** |

Data are presented as means $\pm$ SD or medians (IQR). Statistical differences were calculated by Student's unpaired *t* test or the Mann–Whitney test. Significant *P*-values are presented in bold. oGTT, oral glucose tolerance test; C-section, caesarean section.

subsets of the entire cohort were used (22 and 15 isolations, respectively). For transfection with siRNA, four individual samples of this cohort were used, showing low GWG and, consequently, high *KLRK1-AS1* expression.

## ECFC isolation and culture

Neonatal ECFCs were isolated, characterised and cultured as described previously (Weiss et al., 2022) and frozen in culture medium supplemented with 20% fetal bovine serum (Thermo Fisher Scientific, Waltham, MA, USA) and 10% dimethyl sulfoxide (SERVA, Heidelberg, Germany) before performing experiments. Endothelial Cell Growth Medium MV Kit (PromoCell, Heidelberg, Germany) with 0.1% gentamycin (Thermo Fisher Scientific) was used as culture medium. ECFCs were cultured at 37°C, 21% $O_2$, 5% $CO_2$ in a humidified incubator during experiments and used in passage 5–8.

## Scratch wound healing assay

ECFCs (140,000/ml culture medium per well) were seeded in a 12-well plate pre-coated with 1% porcine skin gelatine (Sigma-Aldrich, St Louis, MO, USA; supplemented with 1% gentamycin (Thermo Fisher Scientific)). After 24 h, the confluent cell monolayer was scratched with a 100 $\mu$l pipette tip. Images were taken every 2 h using an inverted microscope (Olympus CKX53). The width of the scratch was measured at the same three vertical positions per image and the mean width was calculated. The width of the scratch (0 h) and the percentage of decrease of the

scratch (2, 4 and 6 h) were used for statistical analyses. The assay was conducted in triplicate.

## ECIS wound healing assay

Eight-well arrays (8W1E, Ibidi, Munich, Germany) were pre-treated with 10 mM L-cysteine (Sigma-Aldrich) and coated with 1% gelatine (diluted in 150 mM NaCl; Sigma-Aldrich). ECFCs were seeded at a density of 100,000 cells per well and resistance at 4000 Hz was measured using the Electrical Cell-Substrate Impedance Sensing system (ECIS, Applied Biophysics, Troy, NY, USA). After impedance levels have stabilized at 48 h after seeding, a wound was generated by applying a lethal electrical pulse (2600 $\mu$A, 48,000 Hz, 4 s), and the potential of recovery after wounding was measured every hour in relation to the resistance levels just before setting the wound (0 h). The assay was performed in duplicate.

## RNA sequencing

Total RNA was extracted from neonatal ECFCs using the AllPrep DNA/RNA/miRNA Kit (Qiagen, Hilden, Germany) according to the manufacturer's instructions. After determination of suitable RNA quality (RNA integrity number, RIN) scores via the RNA TapeStation system (Agilent, Santa Clara, CA, USA), libraries were prepared with the TruSeq stranded mRNA kit (Illumina, San Diego, CA, USA) by the Victorian Clinical Genetic Services (VCGS) Sequencing Service (Melbourne, Australia). Subsequently, sequencing was performed on the NovaSeq 6000 (Illumina) with 150 bp paired ends. Sequencing reads were aligned to the GRCh37

(hg19) human reference transcriptome using Bowtie (Langmead, 2010). Quantification of gene expression levels as reads per kilobase per million (RPKM) and counts was performed using MMSEQ (Turro et al., 2011). Reads/transcripts were normalized using DESeq2 (Love et al., 2014) and transcripts were annotated as protein coding or lincRNA using Biomart (Smedley et al., 2009). Only the subset of the gene biotype 'lincRNA' was considered for downstream analysis. Differentially expressed transcripts were identified by linear regression using DESeq2 with an unadjusted *P*-value cut-off of <0.05 considered significant.

### siRNA transfection

*KLRK1-AS1* was silenced by siRNA transfection in a subset of four ECFC isolations from pregnancies with low GWG ($7.0 \pm 3.6$ kg) and thus high *KLRK1-AS1* expression ($2.50 \pm 1.87$ RPKM). ECFCs (150,000 per well) were seeded in a six-well plate and left to attach for 24 h. Cells at $60-70\%$ confluency were transfected with 50 nM siRNA (Sigma-Aldrich, sense: GAUCAGAGAAAGAAGCAUA, antisense: UAUGCUUCUUUCUCUGAUC) using 2.25 $\mu$l Lipofectamine RNAiMax Transfection Reagent (Thermo Fisher Scientific) in OptiMEM Reduced Serum Medium, GlutaMAX Supplement (Thermo Fisher Scientific) in a total volume of 1.4 ml per well. Non-targeting siRNA (MISSION siRNA Universal Negative Control #1, Sigma-Aldrich) was transfected as control. Transfection medium was removed after 4 h and replaced with culture medium. Twenty-four hours after transfection, cells were detached and seeded for experiments as well as for RNA extraction to confirm the knockdown.

### RT-qPCR

Validation of RNA sequencing data and knockdown efficiency of the siRNA transfection was determined by RT-qPCR. Total RNA was extracted using the miRNeasy Mini Kit (Qiagen) according to the manufacturer's instructions. cDNA was transcribed with the LunaScript RT SuperMix Kit (New England BioLabs, Ipswich, MA, USA) and qPCR performed with the Luna Universal qPCR Master Mix (New England BioLabs) and the CFX96 Touch Real-Time PCR Detections System (Bio-Rad Laboratories, Hercules, CA, USA). Human primer sequences ($5'-3'$, Sigma-Aldrich) were used to measure *KLRK1-AS1* expression (forward TGAAACGGATTCCCATGGCT, reverse TGCTTCTTTCTCTGATCTGTGTCT) normalised to *RPLP0* (forward TCGACAATGGCAGCATCTAC, reverse ATCCGTCTCCACAGACAAGG) to obtain $\Delta C_\text{t}$ values. Samples were analysed in triplicate.

### Statistics

Statistical analysis was performed with GraphPad Prism software, V9 (GraphPad Software, San Diego, CA, USA; RRID:SCR_019096). Normal distribution of data was determined with the Shapiro–Wilk test and QQ plots. Group comparisons for subjects' characteristics were calculated with Student's two-tailed unpaired *t* test or a Mann–Whitney test. Accordingly, data are presented as means with standard deviation (SD) or median with inter-quartile range (IQR). Differences between *KLRK1-AS1* and *KLRK1* expression were calculated by a Wilcoxon test and thus data are presented as median with IQR. Group comparison of functional assays was performed with two-tailed unpaired (effect of GWG) or paired (effect of knockdown) *t* test, as data were normally distributed. Thus, results are presented with mean and SD. Correlations were analysed with Pearson's or Spearman's correlation according to data Gaussian distribution. A *P*-value < 0.05 was considered as statistically significant. The sample size *n* is the number of individual ECFC isolations.

## Results

### Higher GWG reduces wound healing of neonatal ECFCs

To investigate the effect of GWG on neonatal ECFC function, we divided our cohort of primary cord blood-derived ECFC isolations ($n = 22$) into two equal groups, one from pregnancies with a GWG of <13 kg ($n = 11$) and the other from pregnancies with a GWG of >13 kg ($n = 11$). As a key function of ECFCs, we measured wound healing capacity using a scratch assay. After generating a scratch in the confluent ECFC monolayer, we analysed wound healing migration over time (Fig. 1*A*–*H*) and related it to maternal GWG. After scratching (Fig. 1*I*, GWG <13 kg: $522.3 \pm 67.5$ $\mu$m, GWG >13 kg: $512.7 \pm 76.5$ $\mu$m, $P = 0.757$), there was no difference in the healing of the scratch after 2 h (Fig. 1*J*, GWG <13 kg: $13.88 \pm 4.52\%$, GWG >13 kg: $14.34 \pm 8.30\%$, $P = 0.873$) and 4 h (Fig. 1*K*, GWG <13 kg: $29.44 \pm 8.83\%$, GWG >13 kg: $24.90 \pm 10.65\%$, $P = 0.290$). After 6 h, wound healing was reduced by 25% in neonatal ECFCs obtained after pregnancies with higher GWG compared to the ones obtained after pregnancies with lower GWG (Fig. 1*L*, GWG <13 kg: $48.48 \pm 14.48\%$, GWG >13 kg: $36.26 \pm 10.55\%$, $P = 0.041$). Moreover, in all ECFC isolations, wound healing after 6 h correlated inversely with GWG (Fig. 1*M*, $R = -0.538$, $P = 0.012$).

In the next step, we confirmed the observation of slower ECFC wound healing after exposure to higher GWG using 15 neonatal ECFC isolations for a more standardised experimental approach, i.e. based on changes

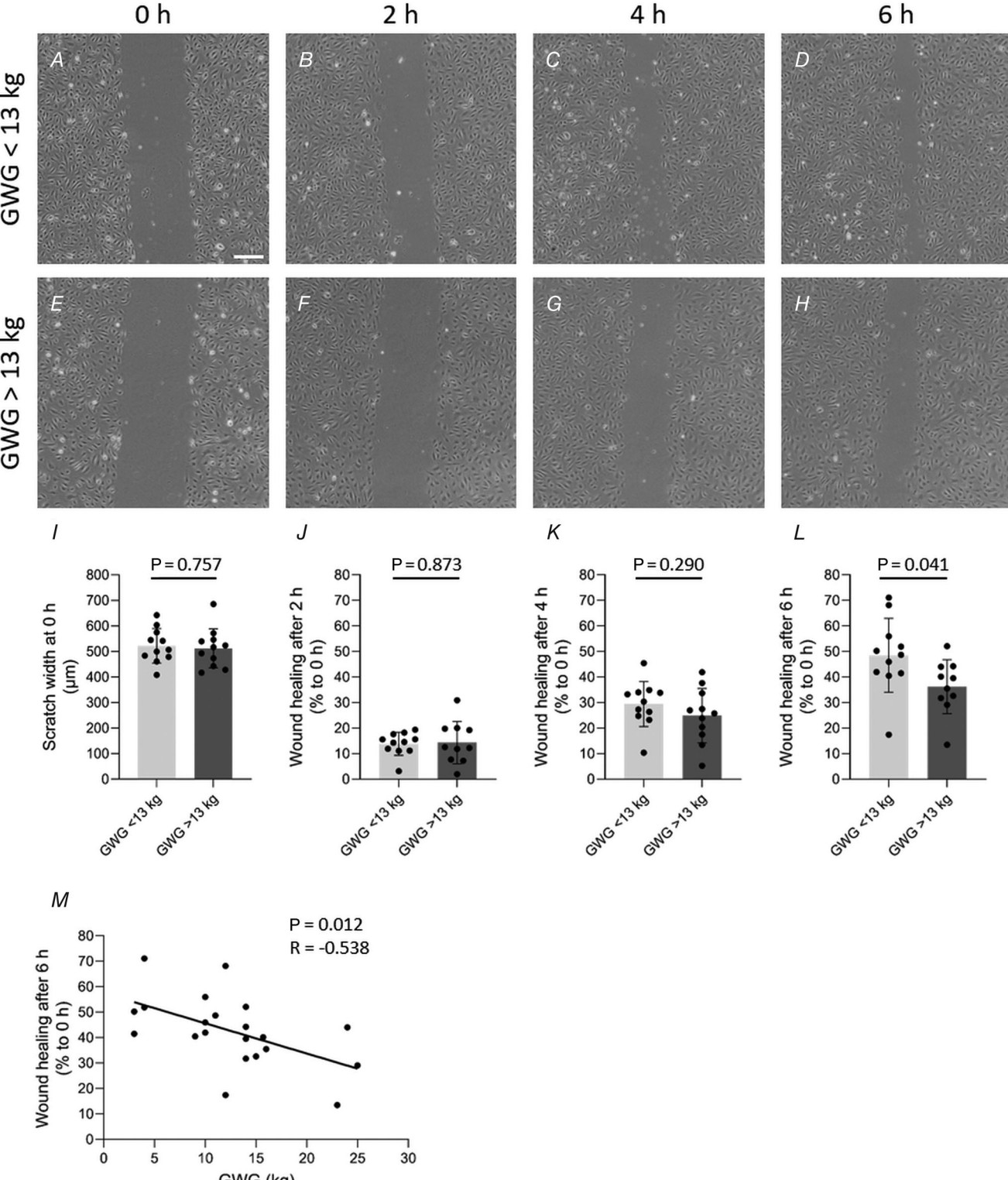

**Figure 1. Effect of maternal gestational weight gain on wound healing capacity of neonatal ECFCs using scratch assay**

ECFC isolated after pregnancies with GWG <13 kg *versus* >13 kg were compared. *A–H*, the confluent ECFC monolayer was scratched, and wound healing was observed over time. *I–L*, the width of the scratch at the beginning (*I*, 0 h) and in relation to it after 2 h (*J*), 4 h (*K*) and 6 h (*L*) was related to maternal gestational weight gain (unpaired *t* test, mean ± SD, *n* = 22). *M*, wound healing after 6 h correlated inversely with maternal gestational weight gain (Pearson, *n* = 22). Scale bar: 200 $\mu$m. GWG, gestational weight gain.

in monolayer impedance using an ECIS device. Therefore, after ECFCs reached confluency characterised by stable impedance (capacitance) against the applied current, the cell monolayer was wounded by lethal electroporation, and the impedance dropped. The following increase in monolayer resistance was observed over time, representing wound healing capacity and recovery of barrier function (Fig. 2*A*). Interestingly, wounding disturbed monolayer impedance of the higher GWG group with 11% lower barrier function at 20 h when compared to the lower GWG group (Fig. 2*B*, GWG <13 kg: 1.03 ± 0.08, GWG >13 kg: 0.91 ± 0.07, $P = 0.014$). Forty hours after wounding, the delayed barrier recovery had caught up and resistance levels were again similar between the two groups (Fig. 2*C*, GWG <13 kg: 0.97 ± 0.22, GWG >13 kg: 0.93 ± 0.24, $P = 0.730$).

### Higher GWG reduces expression of lncRNA *KLRK1-AS1* in neonatal ECFCs

A growing body of evidence highlights the role of lncRNAs in the regulation of endothelial function (Hofmann et al., 2019; Jae & Dimmeler, 2020; Kremer, Bink et al., 2022; Michalik et al., 2014; Pham et al., 2020; Stanicek et al., 2020). In order to identify a potential contribution of lncRNAs in the slower wound healing and barrier recovery of ECFCs exposed to higher GWG, we compared lncRNA expression based on RNA sequencing data in relation to GWG. For this analysis, we expanded our cohort to 35 subjects and investigated lncRNA

expression based on RNA sequencing data from our group (Gene Expression Omnibus repository with the accession number GSE228990; only donors with information about GWG and without developing gestational diabetes were included). Indeed, we identified lncRNAs that correlated with maternal GWG (Table 2).

The most significant correlation with GWG (log$_2$ fold change (FC) = −0.135, $P = 0.0004$) was identified for the lncRNA *KLRK1-AS1* (killer cell lectin-like receptor K1 antisense RNA; also termed *RP11-277P12.20*, ENSG00000245648, NR_120430, LOC101928100, TP53LC04), an antisense lncRNA overlapping with the protein coding gene *KLRK1* (killer cell lectin-like receptor K1) on the antisense strand and located on chromosome 12 (https://genome.ucsc.edu/). A comparison of *KLRK1-AS1* expression with the gene expression of its host gene *KLRK1* (RNA sequencing) is shown in Fig. 3*A* (*KLRK1-AS1*: 0.44 (0.20, 0.75) RPKM, *KLRK1*: 0.03 (0.01, 0.09) RPKM, $P < 0.001$).

Validation of RNA sequencing data was performed via RT-qPCR and showed again an inverse correlation between GWG and *KLRK1-AS1* expression in neonatal ECFCs (Fig. 3*B*, $R = −0.375$, $P = 0.027$).

### Silencing of *KLRK1-AS1* reduces wound healing of neonatal ECFCs

To assess a potential role of *KLRK1-AS1* in neonatal ECFC function, we silenced *KLRK1-AS1* using a specifically designed siRNA in four individual ECFC isolations from pregnancies with low GWG that showed high *KLRK1-AS1*

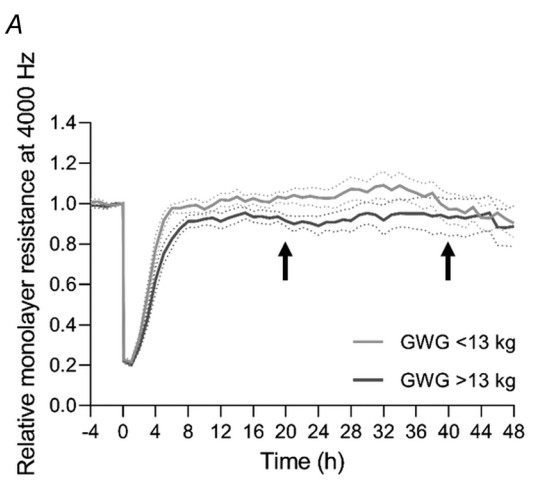
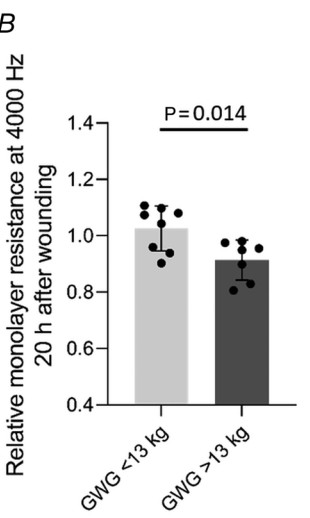
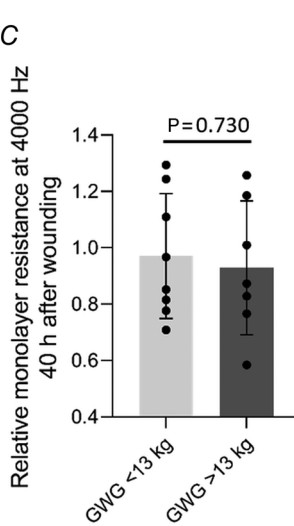

**Figure 2. Effect of maternal gestational weight gain on wound healing capacity of neonatal ECFCs measured via ECIS**
*A*, the confluent ECFC monolayer was wounded by applying high voltage, and wound healing was observed over time every hour in relation to the resistance levels at the time point just before setting the wound (0 h) (mean ± SEM). *B* and *C*, relative levels of resistance related to maternal gestational weight gain are shown for 20 h (*B*) and 40 h (*C*) after wounding (unpaired *t* test, mean ± SD, *n* = 15). GWG, gestational weight gain.

expression. Efficiency of the knockdown by 84% was confirmed through RT-qPCR Fig. 4*A* (si$_{ctrl}$: 100.00 ± 54.12%, si$_{KLRK1-AS1}$: 15.63 ± 9.60%, *P* = 0.007). Of note and similar to the observation of ECFCs after pregnancies with higher GWG and low *KLRK1-AS1* levels, *KLRK1-AS1* knockdown decreased wound healing of neonatal ECFCs (Fig. 4*B*) by 21% 20 h after wounding (Fig. 4*C*, si$_{ctrl}$: 1.05 ± 0.15, si$_{KLRK1-AS1}$: 0.83 ± 0.07, *P* = 0.029). After 40 h, the resistance levels were again similar (Fig. 4*D*, si$_{ctrl}$: 0.92 ± 0.13, si$_{KLRK1-AS1}$: 0.88 ± 0.15, *P* = 0.688).

### Decreased *KLRK1-AS1* expression associates with reduced wound healing in neonatal ECFCs

Finally, we reanalysed the data from the wound healing assays in relation to the individual *KLRK1-AS1* expression of each ECFC donor. Indeed, there was a direct correlation between wound healing capacity with *KLRK1-AS1* expression, both for the scratch assay (Fig. 5*A*, *R* = 0.447, *P* = 0.033) and the impedance-based ECIS assay (Fig. 5*B*, *R* = 0.629, *P* = 014).

### Discussion

We here reveal that maternal GWG affects barrier recovery of isolated neonatal ECFCs after injury. Specifically, higher GWG prolongs the time required for closing the monolayer wound and for reestablishment of barrier impedance, one of the main features of ECFCs. In order to determine whether this disturbed ECFC function involves the action of lncRNAs, we analysed the expression of lncRNAs dependent on GWG. Thus, we identified *KLRK1-AS1* as a positive regulator of ECFC wound healing and barrier recovery, whose expression is attenuated with increasing maternal GWG.

Indisputably, GWG not only impacts maternal health but also fetal development, and it can even affect the offspring's health in the long-term. Particularly low or high GWG elevates the incidence of adverse maternal and offspring outcomes (Goldstein et al., 2017). Several studies showed a direct association between GWG and growth of the fetus (Kominiarek & Peaceman, 2017) with high GWG increasing the risk for the fetus being large for gestational age and macrosomia, and low GWG increasing the risk for the fetus being small for gestational age (Goldstein et al., 2017). Both, low (<2500 g) and high (≥4000 g) birth weight offspring have an enhanced risk of developing type 2 diabetes in childhood, showing a U-shaped association (Wei et al., 2003). Similarly, the effect of high *versus* low GWG on cardiovascular health in childhood follows a U-shaped association (Fraser et al., 2010). In our study, we did not observe a U-shaped effect on ECFC function. However, we have to highlight that our cohort did not include many extremes of GWG. Only four subjects gained less than 5 kg and five subjects more than 20 kg of weight during pregnancy, which may hide any possible U-shaped associations.

lncRNAs are the most prevalent type of non-mRNA transcripts with no or low protein-coding capacity (Ma et al., 2013). The human genome encodes about 100,000 lncRNAs, but a function has only been identified for <5% of them. The underlying reason is their minor conservation between species and their functional heterogeneity (Juni et al., 2022). This functional versatility of lncRNAs is reflected by the different levels at which they can modulate gene expression: as lncRNAs can interact with DNA, RNA and chromatin, they can regulate gene transcription on a transcriptional, post-transcriptional or epigenetic level (Cui et al., 2021).

Recent literature has investigated the role of lncRNAs in endothelial function and identified lncRNA *MEG8*

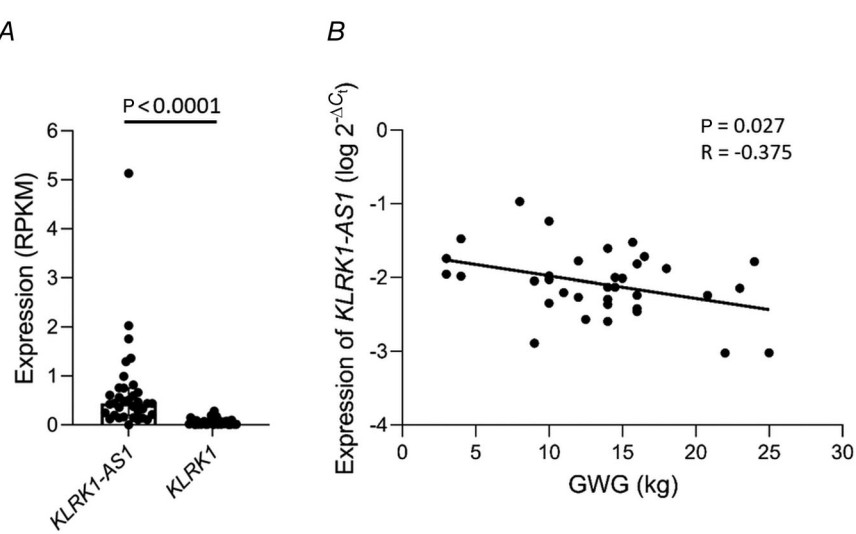

**Figure 3. Expression of lncRNA *KLRK1-AS1* in neonatal ECFCs is independent of *KLRK1* and related to maternal gestational weight gain**
*A*, the expression of lncRNA *KLRK1-AS1* and the gene with which it overlaps on the antisense strand, *KLRK1*, was analysed using RNA sequencing (Wilcoxon test, median with IQR, *n* = 35). *B*, increasing gestational weight gain decreases the expression of *KLRK1-AS1* in neonatal ECFCs (Pearson with Δ$C_t$, *n* = 35). Graph represents logarithmic $2^{-\Delta C_t}$ values. GWG, gestational weigh gain.

as a positive regulator of angiogenic sprouting and proliferation (Kremer, Bink et al., 2022; Kremer, Stanicek et al., 2022). Also, lncRNA *Aerrie* beneficially controls angiogenesis, migration and barrier integrity, and is an important regulator of DNA damage repair (Pham et al., 2020). LncRNA *Lassie* has a positive effect on survival and barrier function (Stanicek et al., 2020), while lncRNA

*H19* protects from inflammation and cellular senescence and promotes monocyte adhesion (Hofmann et al., 2019). Similarly, lncRNA *MALAT1* participates in vascular inflammation, inhibiting migration and angiogenic sprouting, but stimulating proliferation (Michalik et al., 2014). These findings point to an important role in the regulation of endothelial cell function, endothelial

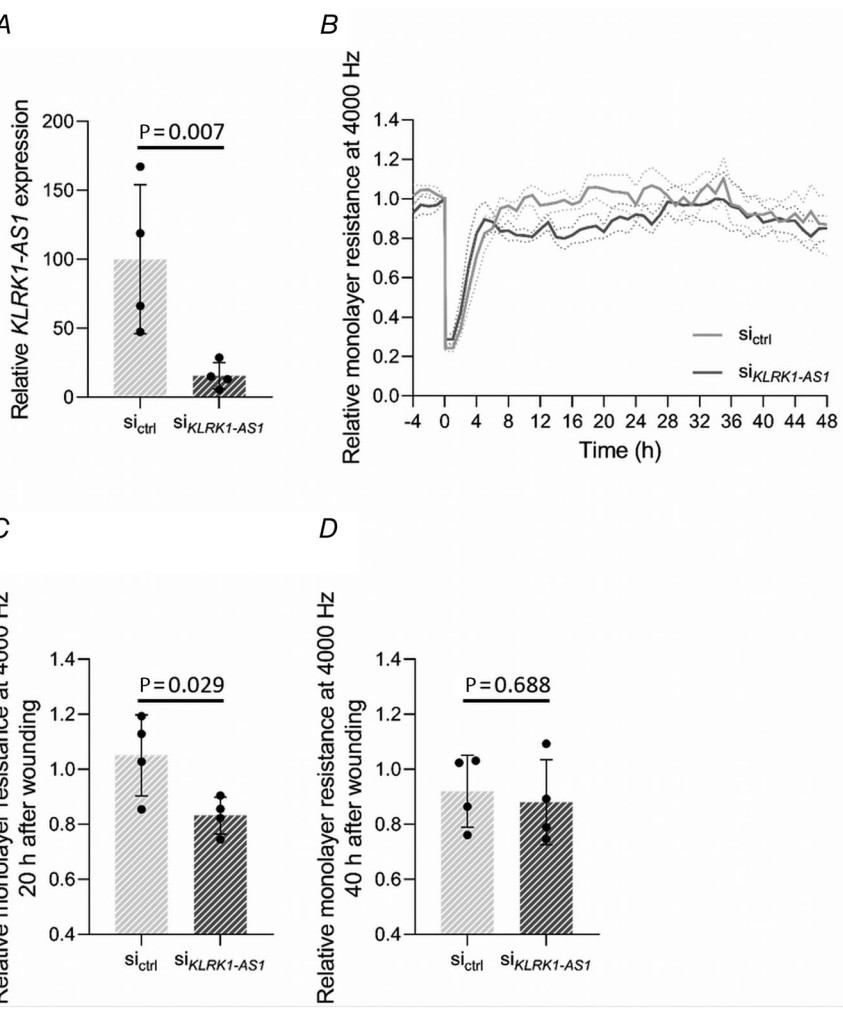

**Figure 4. *KLRK1-AS1* knockdown delays wound healing of neonatal ECFC**
*A*, the expression of *KLRK1-AS1* was downregulated with a specifically designed siRNA (paired *t* test with $\Delta C_t$ values) in four individual ECFC isolations of pregnancies with low gestational weight gain and revealing high *KLRK1-AS1* expression. The graph represents FC *versus* control siRNA, calculated via the $2^{-\Delta C_t}$ method (mean ± SD). *B*, the confluent ECFC monolayer with and without silencing of *KLRK1-AS1* was wounded by applying high voltage, and wound healing was observed over time every hour in relation to the resistance levels at the time point just before setting the wound (0 h) (mean ± SEM). *C* and *D*, relative levels of resistance related to the knockdown of *KLRK1-AS1* are shown for 20 h (*C*) and 40 h (*D*) after wounding (paired *t* test, mean ± SD, *n* = 4, in duplicates). Same ECFC isolations were transfected with non-targeting siRNA (si$_{ctrl}$) or siRNA against *KLRK1-AS1* (si$_{KLRK1-AS1}$).

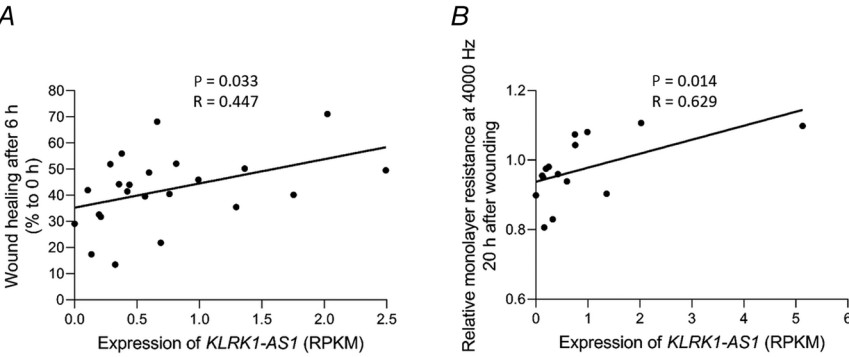

**Figure 5. Wound healing of neonatal ECFC correlates with the expression of *KLRK1-AS1***
Wound healing measured via scratch assay (*A*, *n* = 22) and via ECIS (*B*, *n* = 15) revealed a direct correlation with the expression of *KLRK1-AS1* (Spearman).

**Table 2. Correlations of maternal GWG with the expression of lncRNAs in fetal ECFC sorted by *P*-value**

| Ensemble ID | Name | log$_2$FC | *P* |
|---|---|---|---|
| ENSG00000245648 | RP11-277P12.20 | −0.135 | <0.001 |
| ENSG00000250786 | CTD-2001E22.2 | 0.126 | 0.003 |
| ENSG00000234690 | AC073283.4 | 0.093 | 0.003 |
| ENSG00000235385 | GS1-600G8.5 | −0.124 | 0.005 |
| ENSG00000253522 | hsa-mir-146a | −0.097 | 0.006 |
| ENSG00000130600 | H19 | −0.116 | 0.008 |
| ENSG00000234883 | MIR155HG | 0.031 | 0.010 |
| ENSG00000248187 | RP11-184M15.1 | 0.117 | 0.010 |
| ENSG00000237357 | RP11-475I24.3 | 0.070 | 0.013 |
| ENSG00000232063 | RP11-307E17.8 | 0.045 | 0.016 |
| ENSG00000237945 | AP000569.8 | 0.084 | 0.017 |
| ENSG00000232677 | AC092296.1 | 0.044 | 0.019 |
| ENSG00000228624 | RP3-399L15.3 | −0.065 | 0.021 |
| ENSG00000246523 | RP11-736K20.6 | −0.051 | 0.023 |
| ENSG00000231210 | AC006159.3 | −0.143 | 0.025 |
| ENSG00000228649 | AC005682.5 | 0.041 | 0.027 |
| ENSG00000260396 | AC012065.7 | −0.109 | 0.028 |
| ENSG00000264112 | RP11-159D12.2 | 0.017 | 0.028 |
| ENSG00000259313 | CTA-313A17.5 | −0.059 | 0.028 |
| ENSG00000266753 | RP11-690G19.3 | −0.035 | 0.030 |
| ENSG00000182165 | TP53TG1 | −0.032 | 0.032 |
| ENSG00000231632 | RP11-82L18.4 | 0.088 | 0.033 |
| ENSG00000264247 | RP11-231E4.4 | −0.018 | 0.034 |
| ENSG00000214870 | AC004540.5 | −0.056 | 0.034 |
| ENSG00000259959 | RP11-121C2.2 | 0.022 | 0.036 |
| ENSG00000248202 | RP11-455B3.1 | 0.081 | 0.038 |
| ENSG00000247708 | RP11-323F5.2 | 0.073 | 0.039 |
| ENSG00000223380 | SEC22B | 0.012 | 0.039 |
| ENSG00000226752 | RP11-27I1.2 | −0.031 | 0.040 |
| ENSG00000228925 | AC016722.4 | 0.042 | 0.047 |
| ENSG00000224000 | RP3-471M13.2 | 0.077 | 0.048 |
| ENSG00000232018 | AL132709.8 | 0.042 | 0.048 |

For analysis, the RPKM of the individual isolations were used. FC, fold change.

dysfunction and CVD (Bink et al., 2019; Uchida & Dimmeler, 2015).

We here observed an effect of GWG on endothelial wound healing via *KLRK1-AS1*. Whilst, to the best of our knowledge, this is the first report investigating the effect of GWG on fetal endothelial function, it has already been shown that also gestational diabetes alters wound healing of umbilical cord endothelial cells (HUVECs) (Ye et al., 2018). Interestingly, also here, a lncRNA, i.e. *MEG3*, is involved. In fact, the role of lncRNAs in endothelial dysfunction associated with metabolic derangements is not surprising due to their involvement in inflammation and oxidative stress (Li et al., 2022; Pant et al., 2023; Zhou et al., 2020).

Since lncRNAs are still understudied, little is known about their role in fetal programming. Elevated expression of lncRNA *MEG3* induces endothelial dysfunction in HUVECs of IVF-born offspring (Jiang et al., 2021). In rats, overnutrition of pregnant dams alters hepatic lncRNAs in male offspring (Zhang et al., 2021). Moreover, accumulating evidence suggests that lncRNAs are involved in the regulation of placental development and function (McAninch et al., 2017), which may have the potential to influence fetal development and program the fetus for diseases.

Only a few studies exist investigating the function of *KLRK1-AS1*, and a role in endothelial function has not been described before. *KLRK1-AS1* is, in the anti-sense direction, embedded in the protein-coding gene *KLRK1*, a protein expressed on natural killer cells (https://www.genecards.org/cgi-bin/carddisp.pl?gene=KLRK1&keywords=KLRK1). However, based on the RNA sequencing data of our cohort, *KLRK1* expression in ECFCs is very weak (Fig. 3*A*) when compared to *KLRK1-AS1*, and precludes joint control of gene expression. Within the human body, *KLRK1-AS1* shows its highest expression in spleen and lymph node (https://www.ncbi.nlm.nih.gov/gene/101928100). *KLRK1-AS1* is upregulated in neuroblastoma patients with possible involvement in differentiation and maturation of neurons (Meng et al., 2020). A recent publication has revealed that *KLRK1-AS1* is upregulated by TP53 and encodes a peptide (TP53LC04), which affects cell proliferation *in vitro* (Xu et al., 2022). Interestingly, also another TP53-regulated non-coding RNA was negatively correlated with maternal GWG in ECFCs (Table 2), i.e. *TP53TG1*. In fact, TP53 activates various cellular signalling pathways associated with metabolic functions, but the relationship of TP53 with the overall metabolism is not well understood (Lacroix et al., 2020). Besides the regulation by TP53, a connection of *KLRK1-AS1* expression with inflammatory conditions was identified: non-smoking patients with chronic obstructive pulmonary disease have reduced expression of *KLRK1-AS1* (Qian et al., 2018) and, concomitantly, elevated levels of C-reactive protein (CRP) (Dahl et al., 2011). Also, women with excessive GWG have increased levels of CRP during pregnancy (Hrolfsdottir et al., 2016), and neonatal CRP levels correlated with the maternal levels (Raguz et al., 2016). Hence, as a hypothesis, lncRNA *KLRK1-AS1* may be downregulated under inflammatory conditions, suggesting a subtle pro-inflammatory intrauterine environment in mothers with higher GWG that upregulates *KLRK1-AS1* in neonatal ECFCs. Notably, there is also a link between TP53 and inflammation (Gudkov et al., 2011).

The question that naturally arises is what the increased expression of *KLRK1-AS1* or the delayed wound healing and barrier recovery of neonatal circulating ECFCs mean for the neonates of pregnancies with higher GWG. We have here focused on the function of wound healing as a representative process, as this is one of the key

physiological functions of circulating ECFCs. In the scratch assay, we observed that at the end of the healing process, i.e. at 6 h, the closure of the gap was delayed in the higher GWG group. In the ECIS assay, we also observed a long-term effect on the monolayer. Thus, we believe that contact in between the closing cells, or the establishment of cell–cell or cell–matrix connections may be delayed in ECFCs exposed to higher GWG. Nonetheless, the key message of our study is the finding that increased maternal GWG affects the function of neonatal endothelial cells.

Cord blood-derived ECFCs are circulating progenitor cells which are present in neonatal blood at high concentrations. Therefore, these cells are likely recruited and incorporated into the vasculature during early neonatal development, where they continue to proliferate. In this way, impaired endothelial cell function of cord blood-derived ECFCs will result in altered neonatal endothelial cell function. Unfortunately, we have not found any literature on whether or how a reduced number or function of ECFCs in umbilical cord blood may affect the neonatal health. However, a growing body of evidence based on studies in animals and epidemiological studies in humans indicates that excessive GWG increases the cardiovascular risk of the offspring in the long term (Poston, 2011; Tam et al., 2018) and the first step of endothelial dysfunction may be primed *in utero*.

Besides *KLRK1-AS1*, we have also found other lncRNAs that correlate with maternal GWG. We hypothesize that these lncRNAs will also affect ECFCs, and functions other than wound healing of the monolayer may be involved. However, because *KLRK1-AS1* showed the strongest correlation with GWG, and because wound healing is an important function of ECFCs, we focused on them.

Our study has several limitations. First, the relatively small sample size may limit the generalizability of the findings. We see it as a limitation that we only know the weight gain during the entire pregnancy, but not within the individual trimesters. For fetal development and growth, it is likely to make a difference whether the greatest weight gain occurs in early or in late pregnancy (Galjaard et al., 2013). Furthermore, we are aware of the fact that the sequencing method used only detected lncRNAs containing a $3'$poly(A) tail similar to protein-coding mRNAs, while several lncRNAs are lacking this feature (Wilusz, 2016) and were thus not detected in our RNA sequencing analysis. Finally, we are aware that our results are a snapshot at the time of birth and that we cannot make any statements about the long-term consequences for the vascular health in the offspring. However, since in this study we did not examine placental endothelial cells or HUVECs, but endothelial cells from umbilical cord blood that circulate and act in the neonate and, upon recruitment, become part of the neonatal endothelium, we conclude that higher GWG leads to changes in the endothelial cell function of the neonate. How long these changes will last and whether this could have an impact on the health of the child, we can only speculate, and long-term studies are important to investigate the effect of higher GWG on the offspring's risk of CVD in later life.

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

## Additional information

### Data availability statement

RNA sequencing data are deposited in the Gene Expression Omnibus repository with the accession number GSE228990.

### Competing interests

The authors declare that they have no conflict of interests.

### Author contributions

Experiments were performed at the Perinatal Research Laboratory, Department of Obstetrics and Gynaecology, Medical University of Graz, Graz, Austria, and the Department of Physiology, Amsterdam University Medical Centers, Vrije Universiteit Amsterdam, Amsterdam, the Netherlands. RNA sequencing was performed at the Molecular Immunity, Infection and Immunity Theme, Murdoch Children's Research Institute, Parkville, VIC, Australia. E.W., U.H. and R.A.B. conceptualised and designed the study. E.W., A.S., C.T., M.S.D., A.S.vB. and A.K.S. executed experiments. E.W., A.S., B.N., R.S. and U.H. analysed data. E.W. and U.H. were responsible for manuscript drafting, which was critically discussed by R.A.B. and B.N. All authors revised and approved the manuscript. All authors agree to be accountable for all aspects of the work in ensuring that questions related to the accuracy or integrity of any part of the work are appropriately investigated and resolved. All persons designated as authors qualify for authorship, and all those who qualify for authorship are listed.

### Funding

E.W. was supported by the Austrian Science Fund FWF (DOC 31-B26) and the Medical University of Graz, Austria, through the PhD Program Inflammatory Disorders in Pregnancy (DP-iDP). E.W. and A.K.S. were financed by the FWF project KLI 1023 (to U.H.). C.T. and M.S.D. were funded by the Portuguese Foundation for Science and Technology under the FCT-doctoral Fellowships: C.T., SFRH/BD/11924/2022; M.S.D., SFRH/BD/11934/2022.

### Acknowledgements

The authors thank Bettina Amtmann and Petra Winkler for collection of umbilical cord blood.

### Keywords

endothelial colony forming cells (ECFC), fetal programming, gestational weight gain, lncRNA (long non-coding RNA), wound healing

## Supporting information

Additional supporting information can be found online in the Supporting Information section at the end of the HTML view of the article. Supporting information files available:

**Statistical Summary Document**
**Peer Review History**

