## [Peer Review History · The Journal of Physiology]

Higher gestational weight gain delays wound healing and reduces expression of long non-coding RNA KLRK1-AS1 in neonatal endothelial progenitor cells

Elisa Weiss, Anna Schrüfer, Carolina Tocantins, Mariana Simoes Diniz, Boris Novakovic, Anke S van Bergen, Azra Kulovic-Sissawo, Richard Saffery, Reinier A Boon, and Ursula Hiden

DOI: 10.1113/JP284871

Corresponding author(s): Ursula Hiden (ursula.hiden@medunigraz.at)

The following individual(s) involved in review of this submission have agreed to reveal their identity: Sarah J. Chapple (Referee #2)

Review Timeline:

Submission Date:	12-Apr-2023
Editorial Decision:	18-May-2023
Revision Received:	13-Jun-2023
Editorial Decision:	29-Jun-2023
Revision Received:	29-Jun-2023
Accepted:	03-Jul-2023

Senior Editor: Laura Bennet

Reviewing Editor: Rebecca Simmons

Transaction Report:

Dear Ms Hiden,

Re: JP-RP-2023-284871 "Higher gestational weight gain delays wound healing and reduces expression of long non-coding RNA KLRK1-AS1 in neonatal endothelial progenitor cells" by Elisa Weiss, Anna Schrüfer, Carolina Tocantins, Mariana Simoes Diniz, Boris Novakovic, Anke S van Bergen, Azra Kulovic-Sissawo, Richard Saffery, Reinier A Boon, and Ursula Hiden

Thank you for submitting your manuscript to The Journal of Physiology. It has been assessed by a Reviewing Editor and by 2 expert referees and we are pleased to tell you that it is acceptable for publication following satisfactory revision.

REVISION CHECKLIST:

We look forward to receiving your revised submission.

Yours sincerely,

Professor Laura Bennet
Senior Editor
The Journal of Physiology
<https://jp.msubmit.net>
<http://jp.physoc.org>
The Physiological Society
Hodgkin Huxley House
30 Farringdon Lane
London, EC1R 3AW
UK
<http://www.physoc.org>
<http://journals.physoc.org>

REQUIRED ITEMS

-Include a Key Points list in the article itself, before the Abstract.

-Author photo and profile. First (or joint first) authors are asked to provide a short biography (no more than 100 words for one author or 150 words in total for joint first authors) and a portrait photograph. These should be uploaded and clearly labelled with the revised version of the manuscript. See Information for Authors for further details.

-You must start the Methods section with a paragraph headed Ethical Approval. If experiments were conducted on humans confirmation that informed consent was obtained, preferably in writing, that the studies conformed to the standards set by the latest revision of the Declaration of Helsinki, and that the procedures were approved by a properly constituted ethics committee, which should be named, must be included in the article file. If the research study was registered (clause 35 of the Declaration of Helsinki) the registration database should be indicated, otherwise the lack of registration should be noted as an exception (e.g. The study conformed to the standards set by the Declaration of Helsinki, except for registration in a database.). For further information see: <https://physoc.onlinelibrary.wiley.com/hub/human-experiments>

-The Reference List must be in Journal format

-Your manuscript must include a complete Additional Information section

-Please upload separate high-quality figure files via the submission form.

-A Statistical Summary Document, summarising the statistics presented in the manuscript, is required upon revision. It must be on the Journal's template, which can be downloaded from the link in the Statistical Summary Document section here: https://jp.msubmit.net/cgi-bin/main.plex?form_type=display_requirements#statistics

-A Data Availability Statement is required for all papers reporting original data. This must be in the Additional Information section of the manuscript itself. It must have the paragraph heading "Data Availability Statement". All data supporting the

results in the paper must be either: in the paper itself; uploaded as Supporting Information for Online Publication; or archived in an appropriate public repository. The statement needs to describe the availability or the absence of shared data. Authors must include in their Statement: a link to the repository they have used, or a statement that it is available as Supporting Information; reference the data in the appropriate sections(s) of their manuscript; and cite the data they have shared in the References section. Whenever possible the scripts and other artefacts used to generate the analyses presented in the paper should also be publicly archived. If sharing data compromises ethical standards or legal requirements then authors are not expected to share it, but must note this in their Statement. For more information, see our Statistics Policy.

-Please include an Abstract Figure file, as well as the figure legend text within the main article file. The Abstract Figure is a piece of artwork designed to give readers an immediate understanding of the research and should summarise the main conclusions. If possible, the image should be easily 'readable' from left to right or top to bottom. It should show the physiological relevance of the manuscript so readers can assess the importance and content of its findings. Abstract Figures should not merely recapitulate other figures in the manuscript. Please try to keep the diagram as simple as possible and without superfluous information that may distract from the main conclusion(s). Abstract Figures must be provided by authors no later than the revised manuscript stage and should be uploaded as a separate file during online submission labelled as File Type 'Abstract Figure'. Please ensure that you include the figure legend in the main article file. All Abstract Figures should be created using BioRender. Authors should use The Journal's premium BioRender account to export high-resolution images. Details on how to use and access the premium account are included as part of this email.

EDITOR COMMENTS

Reviewing Editor:

Please address the comments of the Reviewers.

While reviewer 2 asked for additional studies, I feel that these are out of the scope of the study. Additional limitations that were brought up by the reviewers can be addressed in the Discussion.

REFEREE COMMENTS

Referee #1:

Comments.

- The study investigated the impact of maternal gestational weight gain (GWG) on the function of endothelial colony-forming cells (ECFCs), which are critical for vascular repair and barrier function. The researchers found that higher GWG prolongs the time required for ECFCs to recover after injury and identified a lncRNA, KLRK1-AS1, as a positive regulator of ECFC wound healing and barrier recovery, whose expression attenuates with increasing maternal GWG.
- The significance of this study lies in the fact that it sheds light on a potential mechanism by which maternal GWG can affect fetal development and programming. It suggests that maternal factors, such as GWG, can have a lasting impact on the offspring's health by altering the function of critical cells involved in vascular repair and barrier function. This finding could have implications for interventions aimed at improving maternal health during pregnancy and preventing long-term health problems in the offspring.
- One of the strengths of the study is its use of a well-characterized cohort of pregnant women with a wide range of GWG. The researchers also used advanced techniques, such as RNA sequencing, to identify a potential mechanism underlying the observed effects of GWG on ECFC function. The study's findings could provide important insights into the role of lncRNAs in endothelial function and fetal programming.
- One limitation of the study is its relatively small sample size, which may limit the generalizability of the findings. Additionally, the study did not investigate the potential mechanisms underlying the observed effects of KLRK1-AS1 on ECFC function, which could be the subject of future research.
- Overall, this study provides important insights into the potential mechanisms underlying the effects of maternal GWG on fetal development and programming. It suggests that lncRNAs, such as KLRK1-AS1, may play a critical role in regulating the function of ECFCs and could be a target for interventions aimed at improving maternal and fetal health.

Referee #2:

This study seeks to demonstrate a direct association between increased GWG and reduced ncRNA KLRK1-AS1 expression leading to reduced wound healing. The data included are well presented and discussed, however the MS could be strengthened by the following:

1. In the methods it is unclear what the source of ECFCs used is and the pregnancy characteristics (i.e GWG) associated. Clarify whether the n=4 in Fig. 4 represents the same biological donor or separate donors
2. Study relies on the silencing of KLRK1-AS1 on wound healing, however this conclusion would be further strengthened by performing studies assessing the impact of KLRK1-AS1 overexpression
3. Your study rightly has a clear focus on GWG, however other adverse pregnancy conditions such as GDM are associated with poor wound healing. If available, comment on any literature documenting ncRNAs such as KLRK1-AS1 to widen the discussion
4. Briefly comment on whether there is an association between birthweight and KLRK1-AS1

END OF COMMENTS

Confidential Review

12-Apr-2023

EDITOR COMMENTS

Reviewing Editor:

Please address the comments of the Reviewers.

While reviewer 2 asked for additional studies, I feel that these are out of the scope of the study. Additional limitations that were brought up by the reviewers can be addressed in the Discussion.

REFeree COMMENTS

Referee #1:

- The study investigated the impact of maternal gestational weight gain (GWG) on the function of endothelial colony-forming cells (ECFCs), which are critical for vascular repair and barrier function. The researchers found that higher GWG prolongs the time required for ECFCs to recover after injury and identified a lncRNA, KLRK1-AS1, as a positive regulator of ECFC wound healing and barrier recovery, whose expression attenuates with increasing maternal GWG.
- The significance of this study lies in the fact that it sheds light on a potential mechanism by which maternal GWG can affect fetal development and programming. It suggests that maternal factors, such as GWG, can have a lasting impact on the offspring's health by altering the function of critical cells involved in vascular repair and barrier function. This finding could have implications for interventions aimed at improving maternal health during pregnancy and preventing long-term health problems in the offspring.
- One of the strengths of the study is its use of a well-characterized cohort of pregnant women with a wide range of GWG. The researchers also used advanced techniques, such as RNA sequencing, to identify a potential mechanism underlying the observed effects of GWG on ECFC function. The study's findings could provide important insights into the role of lncRNAs in endothelial function and fetal programming.
- One limitation of the study is its relatively small sample size, which may limit the generalizability of the findings. Additionally, the study did not investigate the potential mechanisms underlying the observed effects of KLRK1-AS1 on ECFC function, which could be the subject of future research.

A: We now include a statement on the limited sample size in the discussion section. We agree that investigating potential underlying mechanisms, i.e., the transcriptional regulation of KLRK1-AS1 as well as the targets that are regulated by it, would be interesting, but outside of the scope of this study.

- Overall, this study provides important insights into the potential mechanisms underlying the effects of maternal GWG on fetal development and programming. It suggests that lncRNAs, such as KLRK1-AS1, may play a critical role in regulating the function of ECFCs and could be a target for interventions aimed at improving maternal and fetal health.

Referee #2:

This study seeks to demonstrate a direct association between increased GWG and reduced ncRNA *KLRK1-AS1* expression leading to reduced wound healing. The data included are well presented and discussed, however the MS could be strengthened by the following:

1. In the methods it is unclear what the source of ECFCs used is and the pregnancy characteristics (i.e. GWG) associated. Clarify whether the n=4 in Fig. 4 represents the same biological donor or separate donors

A: We thank the reviewer for finding this lack of information.

The 4 individual ECFC isolations used for siRNA are part of the entire cohort and characterized by low GWG and high *KLRK1-AS1* expression. We now clarify the characteristics of these 4 ECFC isolations in the methods sections 'study cohort' and 'siRNA transfection', in the results section 'Silencing of *KLRK1-AS1* reduces wound healing of neonatal ECFC' and in the figure legend of Figure 4. The mean GWG and *KLRK1-AS1* expression levels of these ECFC isolations are stated (as mean +/- SD) in the methods section 'siRNA transfection'. The number of technical replicates (duplicates) used for ECIS is now indicated in the legend of Figure 4.

2. Study relies on the silencing of *KLRK1-AS1* on wound healing, however this conclusion would be further strengthened by performing studies assessing the impact of *KLRK1-AS1* overexpression

A: We are aware, that overexpression experiments would add to the results. Overexpression is not trivial in primary endothelial cells. The only method that works is lentiviral overexpression and this method, inherently, introduces a longer RNA molecule than the endogenous lncRNA (containing for instance the lentiviral 3'LTR). Furthermore, lncRNAs often work *in cis*, meaning that the genomic location they are transcribed from is very important for their function. Lentiviral overexpression induces random integration in the genome, so you lose that aspect. Therefore, siRNA transfection is widely used (*Reprod Sci.* 2023 Feb;30(2):678-689; *PLoS One.* 2022 Sep 29;17(9):e0265160; *J Cell Sci.* 2022 Jun 15;135(12):jcs259671; *Sci Rep.* 2022 Jan 17;12(1):843; *Int J Mol Sci.* 2021 Jul 28;22(15):8088; *Front Cell Dev Biol.* 2021 Jan 11;8:619079; *Commun Biol.* 2020 May 26;3(1):265; *Cardiovasc Res.* 2019 Jan 1;115(1):230-242), and, most importantly, represents a good solution for our research question, since silencing shifted the cells of women with high GWG to the *KLRK1-AS1* expression level of women with low GWG. Therefore, we decided on this solution.

3. Your study rightly has a clear focus on GWG, however other adverse pregnancy conditions such as GDM are associated with poor wound healing. If available, comment on any literature documenting ncRNAs such as *KLRK1-AS1* to widen the discussion

A: We have now included a statement on altered endothelial function and endothelial wound healing in GDM and the involvement of lncRNAs. The following paragraph was included into the discussion:

We here observed an effect of GWG on endothelial wound healing via KLRK1-AS1. Whilst, to the best of our knowledge, this is the first report investigating the effect of GWG on fetal endothelial function, it has already been shown that also gestational diabetes alters wound healing, of umbilical cord endothelial cells (HUVEC) (Ye et al., 2018). Interestingly, also here, a lncRNA, i.e., MEG3, is involved. In fact, the role of lncRNAs in endothelial dysfunction associated with metabolic derangements is not surprising due to their involvement in inflammation and oxidative stress (Zhou et al., 2020; Li et al., 2022; Pant et al., 2023).

4. Briefly comment on whether there is an association between birthweight and KLRK1-AS1.

A: In line with the literature (*Obesity (Silver Spring). 2013 Jul;21(7):1416-22*), neonatal weight/height and placental weight were increased after pregnancies of women with higher GWG (Table 1). However, there was no correlation between neonatal birth weight ($R=-0.091$, $p=0.604$, Spearman, $n=35$), neonatal birth height ($R=-0.105$, $p=0.548$, Spearman, $n=35$) or placental weight ($R=-0.099$, $p=0.579$, Spearman, $n=35$) and *KLRK1-AS1* expression in neonatal ECFC (c.f. graphs below).

Dear Ms Hiden,

Re: JP-RP-2023-284871R1 "Higher gestational weight gain delays wound healing and reduces expression of long non-coding RNA KLRK1-AS1 in neonatal endothelial progenitor cells" by Elisa Weiss, Anna Schrüfer, Carolina Tocantins, Mariana Simoes Diniz, Boris Novakovic, Anke S van Bergen, Azra Kulovic-Sissawo, Richard Saffery, Reinier A Boon, and Ursula Hiden

Thank you for submitting your revised Research Article to The Journal of Physiology. It has been assessed by the original Reviewing Editor and Referees and has been well received. Some final revisions have been requested. Please see the required items section below.

We hope that you will be able to return your revised manuscript within one week. If you require longer than this, please contact journal staff: jp@physoc.org.

We look forward to receiving your revised submission.

Yours sincerely,

Professor Laura Bennet
Senior Editor
The Journal of Physiology
<https://jp.msubmit.net>
<http://jp.physoc.org>
The Physiological Society
Hodgkin Huxley House
30 Farringdon Lane
London, EC1R 3AW
UK
<http://www.physoc.org>
<http://journals.physoc.org>

REQUIRED ITEMS FOR REVISION

-You must start the Methods section with a paragraph headed Ethical Approval. If experiments were conducted on humans confirmation that informed consent was obtained, preferably in writing, that the studies conformed to the standards set by the latest revision of the Declaration of Helsinki, and that the procedures were approved by a properly constituted ethics committee, which should be named, must be included in the article file.

PLEASE NOTE: If the research study was registered (clause 35 of the Declaration of Helsinki) the registration database should be indicated, otherwise the lack of registration should be noted as an exception (e.g. The study conformed to the standards set by the Declaration of Helsinki, except for registration in a database.). For further information see: <https://physoc.onlinelibrary.wiley.com/hub/human-experiments>

-Please include an Abstract Figure legend text within the main article (doc.) file.

EDITOR COMMENTS

Reviewing Editor:

The reviewers issues have been satisfactorily addressed. Congratulations!

REFEREE COMMENTS

Referee #1:

The article is well written and is impactful as the study can open up avenues for further research into interventions aimed at improving neonatal endothelial function. Understanding the specific lncRNAs involved in mediating the effects of GWG on ECFC could potentially lead to the development of targeted therapies

Referee #2:

Comments have been sufficiently addressed. I congratulate the authors on a well reasoned response and well performed study.

END OF COMMENTS

1st Confidential Review

13-Jun-2023

EDITOR COMMENTS

REQUIRED ITEMS FOR REVISION

-You must start the Methods section with a paragraph headed Ethical Approval. If experiments were conducted on humans confirmation that informed consent was obtained, preferably in writing, that the studies conformed to the standards set by the latest revision of the Declaration of Helsinki, and that the procedures were approved by a properly constituted ethics committee, which should be named, must be included in the article file.

PLEASE NOTE: If the research study was registered (clause 35 of the Declaration of Helsinki) the registration database should be indicated, otherwise the lack of registration should be noted as an exception (e.g. The study conformed to the standards set by the Declaration of Helsinki, except for registration in a database.). For further information see: <https://physoc.onlinelibrary.wiley.com/hub/human-experiments>

-Please include an Abstract Figure legend text within the main article (doc.) file.

Answers:

These submission addresses the last final revisions that have been requested:

1) The Ethics Approval has been already included at the beginning of the methods section before, but now we inserted a statement that the study was not registered. It reads:

The study confirmed to the standards set by the Declaration of Helsinki (version 2013), **except for registration in a database**. It was approved by the ethics committee of the Medical University of Graz, Austria (29-319 ex 16/17). Only participants with written informed consent were included.

2) The Abstract Figure Legend was given below the Abstract Figure at the beginning of the manuscript. Now the legend starts with the designation 'Abstract Figure Legend' and reads:

Abstract Figure Legend. Gestational weight gain affects wound healing of fetal endothelial progenitor cells. Cord blood derived endothelial colony forming cells (ECFC) were isolated after pregnancies with high vs low gestational weight gain (GWG) and wound healing capacity was compared. High gestational weight gain caused delayed wound healing of ECFC. Correlation analysis and silencing experiments revealed that wound healing capacity was related with the expression of the long non-coding RNA *KLRK1-AS1*.

We hope that we have adequately addressed both points and that the manuscript now meets also the formal requirements of the journal.

with kind regards,

Ursula Hiden

Dear Dr Hiden,

Re: JP-RP-2023-284871R2 "Higher gestational weight gain delays wound healing and reduces expression of long non-coding RNA KLRK1-AS1 in neonatal endothelial progenitor cells" by Elisa Weiss, Anna Schröder, Carolina Tocantins, Mariana Simoes Diniz, Boris Novakovic, Anke S van Bergen, Azra Kulovic-Sissawo, Richard Saffery, Reinier A Boon, and Ursula Hiden

We are pleased to tell you that your paper has been accepted for publication in The Journal of Physiology.

Authors should note that it is too late at this point to offer corrections prior to proofing. The accepted version will be published online, ahead of the copy edited and typeset version being made available. Major corrections at proof stage, such as changes to figures, will be referred to the Editors for approval before they can be incorporated. Only minor changes, such as to style and consistency, should be made at proof stage. Changes that need to be made after proof stage will usually require a formal correction notice.

Yours sincerely,

Professor Laura Bennet
Senior Editor
The Journal of Physiology
<https://jp.msubmit.net>
<http://jp.physoc.org>
The Physiological Society
Hodgkin Huxley House
30 Farringdon Lane
London, EC1R 3AW
UK
<http://www.physoc.org>
<http://journals.physoc.org>

P.S. - You can help your research get the attention it deserves! Check out Wiley's free Promotion Guide for best-practice recommendations for promoting your work at www.wileyauthors.com/eeo/guide. You can learn more about Wiley Editing Services which offers professional video, design, and writing services to create shareable video abstracts, infographics, conference posters, lay summaries, and research news stories for your research at www.wileyauthors.com/eeo/promotion.

IMPORTANT NOTICE ABOUT OPEN ACCESS: To assist authors whose funding agencies mandate public access to published research findings sooner than 12 months after publication, The Journal of Physiology allows authors to pay an Open Access (OA) fee to have their papers made freely available immediately on publication.

You can check if your funder or institution has a Wiley Open Access Account here: <https://authorservices.wiley.com/author-resources/Journal-Authors/licensing-and-open-access/open-access/author-compliance-tool.html>.